# Paternal DNA methylation is remodeled to maternal levels in rice zygote

Qian Liu [1,3], Xuan Ma [1,3], Xue Li [1], Xinran Zhang [1], Shaoli Zhou[1], Lizhong Xiong [1], Yu Zhao [1] ✉ & Dao-Xiu Zhou [1,2] ✉

Epigenetic reprogramming occurs during reproduction to reset the genome for early development. In flowering plants, mechanistic details of parental methylation remodeling in zygote remain elusive. Here we analyze allele-specific DNA methylation in rice hybrid zygotes and during early embryo development and show that paternal DNA methylation is predominantly remodeled to match maternal allelic levels upon fertilization, which persists after the first zygotic division. The DNA methylation remodeling pattern supports the predominantly maternal-biased gene expression during zygotic genome activation (ZGA) in rice. However, parental allelic-specific methylations are reestablished at the globular embryo stage and associate with allelic-specific histone modification patterns in hybrids. These results reveal that paternal DNA methylation is remodeled to match the maternal pattern during zygotic genome reprogramming and suggest existence of a chromatin memory allowing parental allelic-specific methylation to be maintained in the hybrid.

The zygotic transition, from a fertilized egg to an embryo, is central to animal and plant reproduction. In animals, embryo development depends on maternally provided factors until zygotic genome activation (ZGA) that takes place after one to several cell divisions depending on the species[1]. In flowering plants, ZGA is rapidly initiated and occurs before the first zygotic division[2–5]. Studies in plants have found a large number of de novo expressed genes that are required for zygotic division[6–10]. Parent-of-origin contributions to plant early embryogenesis have been studied at genetic and transcriptomic levels[10]. However, parental contribution to the zygotic transcriptome in plants is under debate[5,11]. Studies in Arabidopsis revealed that maternal transcripts dominate the transcriptomes of embryos at the 2–4 cell and globular stages[12,13], whereas other results indicated that maternal and paternal genomes contribute equally to the transcriptome of early embryos, even at the 1–2 cell stage or elongated zygotes[6,7,14]. However, a reanalysis of the published data[6,7] showed that, on a gene-by-gene basis, the Arabidopsis hybrid (Col/Ler) zygotes do not show equal parental transcriptome contributions; thousands of genes in hybrid zygotes are represented by transcripts from either the maternal or paternal allele, but not both[13]. There is also genetic evidence that maternal alleles of most embryo genes make a more important contribution functional to early embryogenesis than paternal alleles, and that hybridization itself can affect parental genome contributions to early embryogenesis[13,15]. In rice, analysis of allele-specific transcriptome in the zygote revealed that transcription of the zygotic genome is mainly from the maternal alleles, which results in a maternally dominated transcriptome[9]. ZGA is a gradual process that relies on large-scale chromatin reprogramming leading to an increasing number of zygotically expressed genes[1], which may involve crosstalk between the parental epigenomes to control zygote and early development.

In mammals, it is generally assumed that two distinct phases of epigenetic reprogramming serve to prevent inheritance of ancestrally acquired epigenetic signatures. This reprogramming process comprises the erasure of DNA methylation marks from the previous generation followed by a re-establishment of DNA methylation[16]. Unlike in mammals, plant DNA methylation is found to be only partially

---

[1]National Key Laboratory of Crop Genetic Improvement, Hubei Hongshan Laboratory, Huazhong Agricultural University, Wuhan 430070, China. [2]Institute of Plant Science Paris-Saclay (IPS2), CNRS, INRAE, University Paris-Saclay, 91405 Orsay, France. [3]These authors contributed equally: Qian Liu, Xuan Ma. ✉e-mail: zhaoyu@mail.hzau.edu.cn; dao-xiu.zhou@universite-paris-saclay.fr

remodeled or reconfigured in the gametes and the unicellular zygote[17–20]. The partial epigenetic reprogramming of DNA methylation may contribute to stable epigenetic inheritance relatively frequently observed in plants[16]. In the meantime, the DNA methylation remodeling is also essential for plant reproduction, as perturbation of DNA methylation by mutation of DNA demethylase genes affects function of the gametes and impairs the development of zygote and embryo as well as endosperm in rice[20–22]. In plants, knowledge on epigenetic basis and dynamics of the parental contributions during fertilization and early embryogenesis is limited, despite its importance in understanding epigenetic inheritance and the effects of parental genome interactions in the context of non-self pollination in plants.

In this work, we show that in rice hybrid zygotes paternal DNA methylation is remodeled to match the maternal levels, consistent with the predominant maternal transcripts in the zygote transcriptome. Interestingly, the parental allelic or sequence-specific methylations are reestablished at the globular stage of the hybrid embryos and maintained during development. These results reveal a maternal pattern of zygotic epigenome reprogramming in plant and highlight genetic control of parental allelic-specific methylation reestablishment and maintenance in hybrid.

## Results

### Remodeling of the rice gamete methylomes in the zygote upon fertilization

To investigate the parental epigenome dynamics in the zygote, we first analyzed the egg, sperm and zygote (at 6.5 h after pollination, HAP, after the gamete nuclear fusion[9]) DNA methylation patterns of elite hybrid rice "SY63" parental lines (MH63 and ZS97), using a bisulfite sequencing (BS-seq) protocol developed for small numbers of cells[20,23,24]. DNA methylomes data were obtained from 25 eggs or zygotes and 150 sperm cells, two biological replicates were performed with a sequencing depth of about 24.7–75.4 × genome coverage per replicate (Supplementary Table 1). Principal component analysis revealed a high reproducibility of the replicates and a clear difference between the two parental lines (Supplementary Fig. 1a). Boxplots indicated that sperm cells showed globally lower CHG methylation (mCHG) than egg cells in TEs (Supplementary Fig. 2a). Unlike in Arabidopsis sperm where CHH methylation (mCHH) is lost[17], the rice sperm mCHH was higher than the egg level (Supplementary Fig. 1b, c; Supplementary Fig. 2a), which may be due to a different landscape and higher levels of mCHH in the rice genome[25,26]. In the zygote mCG and mCHH levels were lower than in the sperm, while the mCHG was at the intermediate levels of the egg and sperm cells (Supplementary Fig. 2a). Density plots revealed lower mCG in the zygote relative to sperm and egg, and lower mCHH but higher mCHG in the zygote versus sperm (Supplementary Fig. 2b). The analysis confirmed that the parental DNA methylation was rapidly remodeled upon fertilization in rice[20], and suggested a predominant remodeling of the male methylome in the zygote. Scanning differentially methylated regions (DMRs, within 50-bp windows with the cutoff of methylation difference at CG > 0.5, CHG > 0.3, and CHH > 0.1, $P < 0.05$) between the gametes and zygotes revealed that about a third or more of the DMRs concerned non-transposable element (non-TE) regions (Supplementary Fig. 2c). Comparisons between MH63 egg and ZS97 sperm or between ZS97 egg and MH63 sperm, as used in the reciprocal crosses, revealed higher DNA methylation variations than between egg and sperm within the inbred lines (Supplementary Fig. 1c, d).

### A number of given loci tend to be remodeled in the zygote

Next, we analyzed the methylomes of the reciprocal hybrid zygotes (at 6.5 HAP) and globular embryos (GE, at 72 HAP) of SY63 (ZS97 as female, MH63 as male, hereafter referred to as ZM) and MZ (MH63 as female, ZS97 as male) (Supplementary Fig. 1a; Supplementary Table 1). In the hybrid zygotes, the methylation levels appeared higher than in

the male and female gametes, particularly at genic CG and CHG sites (Fig. 1a, b), which was confirmed by density plots (Fig. 1c) and/or DMR scanning (Fig. 1d) and was consistent with previous observations of enhanced DNA methylation in hybrid vegetative tissues[27,28]. Although such reinforcement was not observed in the inbred zygotes (Supplementary Fig. 2a), substantial portions (about 30-66%) of CG and CHG hyper DMRs of the inbred zygote versus sperm (Z – S) or egg (Z – E) (Supplementary Fig. 2c) overlapped with those found in the reciprocal hybrid zygotes (Supplementary Fig. 3a, b), suggesting that DNA methylation at a number of specific loci (Supplementary Fig. 3c, d) tended to be reinforced upon fertilization. Genes with diverse functions were associated the hyper DMRs in the hybrid and inbred zygotes versus sperm (Supplementary Fig. 3c-f; Supplementary Data 1). Most of the genes were lowly expressed or repressed in both sperm and zygotes, while a number of genes were expressed in sperm but repressed in the zygotes (Supplementary Data 1, **labeled in red**). Density plots revealed a clear bimodal distribution pattern of CHH DMR between zygote and sperm (Z – S) or between zygote and egg (Z – E) (Fig. 1c), indicating a fraction of loci showed clearly increased (hyper) or decreased (hypo) methylation at CHH sites in the zygote genome. Further analysis indicated that the hyper methylated CHH sites were enriched in genic regions whereas the hypo-methylated sites were mainly located in TE regions (Supplementary Fig. 4a). Genes with the CHH DMRs were mainly enriched in RNA silencing, defense and developmental pathways (Supplementary Fig. 4b). In the hybrid globular embryos (GE), genic methylation levels were maintained or even augmented compared to the zygote levels, while TE methylation (mainly mCG and mCHG) was lower than in the zygote but close to the gametes or seedling levels (Fig. 1a, b)[29], indicating that DNA methylation continued to be remodeled during early embryogenesis.

### Male genome methylation is remodeled to match the female levels in the zygote

To follow up the egg versus sperm (E – S) DMRs in the zygote, we analyzed their methylation levels in both inbred and hybrid zygotes. In the hybrid zygotes the overall methylation levels of the CG and CHG DMRs between egg and sperm were close to the egg levels, while those of the CHH DMRs paralleled the lower parental levels (Fig. 2a, b). Similar profiles were observed in the inbred zygotes (Supplementary Fig. 5). To confirm the observation, we separated the parental allele-specific reads from the hybrid zygote BS-seq data by using the 1,351,242 single nucleotide polymorphisms (SNPs) between MH63 and ZS97 genomes[30]. The allele-specific methylation reads in the hybrid zygotes were about 10.6–14.1% of the total reads, similar to those observed in DNA methylomes of hybrid rice vegetative tissues[29,31]. In the hybrid zygotes, the methylation levels of both the maternal and paternal alleles of the E – S CG and CHG DMRs (both hypo and hyper) were close to the egg but distinct from the sperm levels (Fig. 2a, b, d, e). To further confirm the results, we crossed the ZH11 variety with MH63 (ZH11 as female, MH63 as male, hereafter referred to as ZHM) and obtained methylation data from 2-cell embryos (harvested at 12 HAP) (Supplementary Table 1). Analysis of parental allele-specific methylation in the 2-cell embryos by using the SNPs between the MH63 and ZH11 genomes[32], obtained a similar result (Fig. 2c). To distinguish between paternal methylation changing to maternal levels from reverting to vegetative levels in the hybrid zygotes, we analyzed the methylation levels of the E – S DMRs in sperm, egg, zygote, shoot and panicles of the paternal lines used to produce the three hybrids (ZM, MZ and ZHM). We observed that the methylation levels of the DMRs in sperm were similar to shoot and panicle of the 3 paternal lines (Supplementary Fig. 6a–c). DNA methylations in egg and sperm of inbred lines are differentially remodeled (Supplementary Fig. 1)[20]. These observations suggested that the paternal alleles of the E – S CG and CHG DMRs were remodeled to match the levels of the maternal alleles rather than to restore to the vegetative levels in the zygote. The data

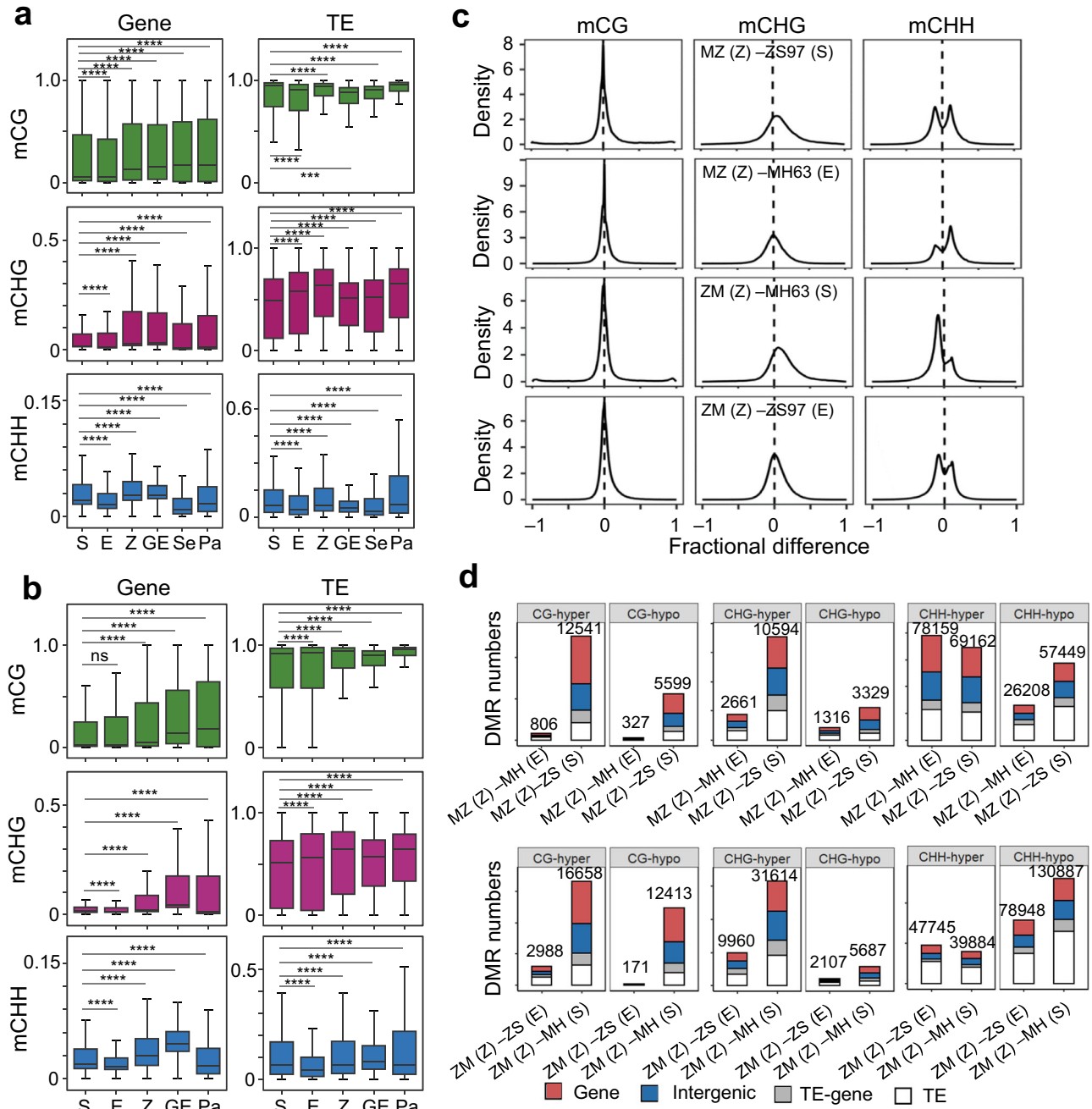

**Fig. 1 | Parental DNA methylation remodeling in the hybrid zygotes.** Boxplots showing TE (number = 375,397) and gene (number = 39,407) CG, CHG, and CHH methylation levels in the hybrid ZM **(a)** zygote (Z), globular embryo (GE), seedling (Se), and panicle (Pa, ZM) compared with MH63 sperm (S) and ZS97 egg (E), and in the hybrid MZ **(b)** zygote (Z), globular embryo (GE), and panicle (Pa) compared with ZS97 sperm (S) and MH63 egg (E). Values of the methylation levels are averages from the two replicates (***P < 0.001, ****P < 0.0001, ns, not significant, two-sided Wilcoxon rank-sum test). The horizontal line within the box represents the median,

box limits represent the interquartile range (IQR), and whiskers represent 1.5 × IQR. **c** Density plot showing the frequency distribution of fractional methylation difference between the reciprocal hybrid (ZM and MZ) zygotes (Z) and the respective sperm (S) and egg (E) cells from ZS97 or MH63. **d** DMR numbers in the hybrid (ZM and MZ) zygote (Z) versus egg (E) or sperm (S) from MH63 (MH) or ZS97 (ZS). Upper panel, MZ, lower panel, ZM. DMRs in gene body, intergenic, TE-gene and TE regions are denoted by red, blue, gray and white, respectively. Source data are provided as a Source Data file.

together indicated that the paternal allele-specific methylation is remodeled to the levels similar to the maternal alleles in the zygote, which persists till at least the 2-cell embryo stage.

**Parental allele-specific methylation was restored during embryogenesis and stably maintained in the hybrids**
To investigate whether the zygotic remodeling of paternal methylation was maintained during embryogenesis, we analyzed the

methylation levels of the E – S DMRs in the GEs of the reciprocal crosses. In the GEs, methylations of the CG and CHG DMRs were at the intermediate levels of the gametes (Fig. 3). However, the levels of CHH DMRs remained to parallel the lower parental levels (Fig. 3), consistent with the observation of mCHH loss during embryogenesis in Arabidopsis[33]. Transcript levels of genes involved in CHH methylation (e.g. *AGO4*, *DCL3*, *DRM2*, and *Pol IV*) were lower in the GE than in the zygote (Supplementary Fig. 7). However, the paternal

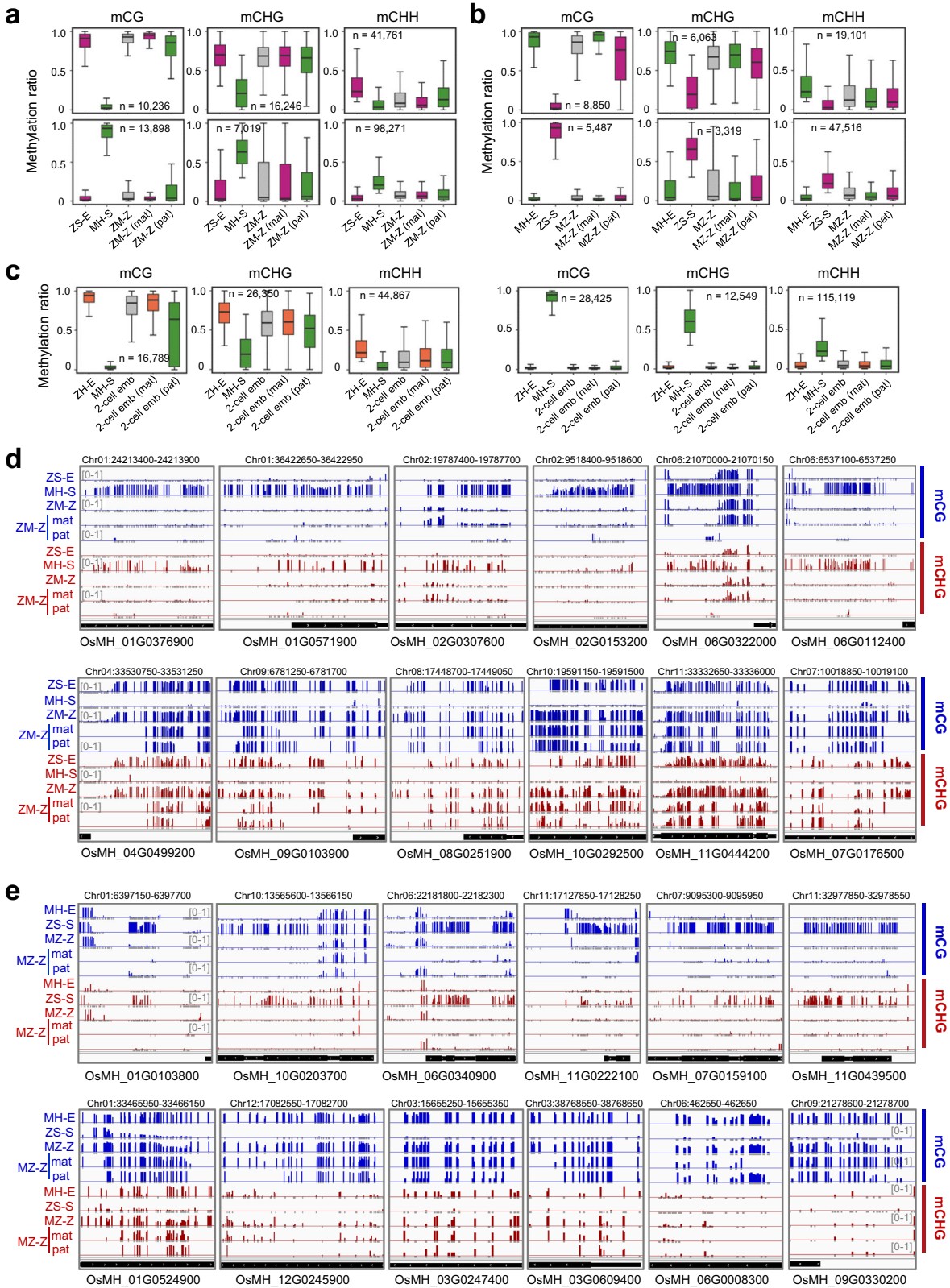

allelic methylations of the CG and CHG DMRs were close to the sperm levels, whereas those of maternal alleles were close to the egg levels (Fig. 3), suggesting that the parental allelic or DNA sequence-specific methylation, which had been observed in seedling and panicle tissues of ZM and MZ[29], and detected in the reciprocal hybrids between NIP and 9311 varieties (Supplementary Fig. 8), are reestablished at the GE stage (Fig. 3).

**Parental methylation difference was associated with distinct histone modifications**

To study whether the reestablishment of parental allelic-specific DNA methylation in the hybrid embryos was related to specific chromatin signatures, we analyzed histone modification marks including H3K27ac, H3K4me3 and H3K9me2 in the E (ZS97) – S (MH63) DMRs using the ChIP-seq data obtained from MH63 and ZS97 seedling

**Fig. 2 | Parental allele-specific methylation in the hybrid zygotes.** Boxplots showing maternal (mat) and paternal (pat) allelic DNA methylation levels of the E – S DMRs in the hybrid zygote of ZM (**a**) and MZ (**b**) compared with the methylation levels of the DMRs in the zygotes (ZM-Z, or MZ-Z) and the respective parental egg (E) and sperm (S) cells from MH63 (MH) or ZS97 (ZS). Upper panel, hyper E – S DMRs; lower panel, E – S hypo DMRs. N denotes the numbers of E – S DMRs. *n* = 2 biologically independent samples for each cell type examined. **c** Maternal (mat) and paternal (pat) allele methylation levels of the E – S DMRs between ZH11 egg (ZH-E) and MH63 sperm (MH-S) in 2-cell embryos of the ZH11 × MH63 hybrid, compared with the levels of the DMRs in sperm (MH-S), egg (ZH-E) and the 2-cell embryos (2-

cell emb). Left panel, hyper E – S DMRs; right panel, E – S hypo DMRs. *n* = 2 biologically independent samples for each cell type examined. Genome browser screenshots of parental allelic CG and CHG methylation levels in ZM (**d**) and MZ (**e**) zygotes compared with the levels in egg, sperm and zygote. CG and CHG methylation are denoted by blue and red, respectively. Upper panel, egg <sperm methylation, lower panel, egg > sperm. Gray bars under the track represent the presence of covered (≥3 reads) cytosine sites in each methylation context. The horizontal line within the box represents the median, box limits represent the interquartile range (IQR), and whiskers represent 1.5 × IQR. Source data are provided as a Source Data file.

tissues[34]. In the CG and CHG hyper DMRs, the active histone marks H3K27ac and H3K4me3 were absent from ZS97, but present at very high levels in MH63 alleles. By contrast, the H3K9me2 (a repressive mark that tightly associates with mCG and mCHG in plants) levels of the DMRs were high in ZS97, but absent from MH63 alleles (Fig. 4a, b; Supplementary Fig. 9a, b). In the hypo DMRs, opposite histone modification profiles were observed (Fig. 4a, b; Supplementary Fig. 9a, b). Similar observations were made for the E (MH63) – S (ZS97) DMRs (Supplementary Fig. 10a, b). Thus, methylation differences between the male and female gametes appeared to associate with distinct histone marks in vegetative tissues of the respective parental lines. To study whether the association could be detected in the hybrid cells, we performed H3K4me3 ChIP-seq of the hybrid ZM seedling tissues, and analyzed the parental allele-specific H3K4me3 by using SNPs between MH63 and ZS97. The analysis revealed that, in E (ZS97) – S (MH63) hyper DMRs, H3K4me3 was depleted from the maternal (ZS97), but present at very high levels in the paternal (MH63) alleles. In the hypo DMRs, a reverse situation was observed (Fig. 4c; Supplementary Fig. 9c; Supplementary Fig. 10c, d). Analysis of chromatin modification data of the reciprocal hybrids between NIP and 9311 varieties[35], revealed a similar result (Supplementary Fig. 11a–c). Together, these data indicated that parental allelic-specific methylation associates with parental allelic-specific histone marks, which may be underlying the reestablishment of parental allelic-specific DNA methylations during early embryogenesis and maintenance during development.

**Parental DNA methylation remodeling mirrors parental contribution to zygotic gene expression**
To study whether the parental methylation remodeling pattern was associated with gene expression in the zygote, using RNA-seq we analyzed transcriptomes of sperm, egg, zygote (6.5 HAP) and GE (72 HAP) of the reciprocal crosses between MH63 and ZS97 (Supplementary Table 2), with 3 biological replicates (r = 0.94 ~ 1.0) (Supplementary Fig. 12a). Principal component analysis indicated that the sperm transcriptomes were distal from those of egg, zygote, and GEs (Supplementary Fig. 12b). Comparison of the hybrid zygotes with the respective egg cells revealed more than 2000 up- and downregulated genes (|log2 (Fold Change)| > 1, Q-value < 0.01) in the reciprocal hybrid zygotes, among which 601 genes were commonly upregulated (Supplementary Fig. 12c, d). These genes are enriched in DNA replication, ethylene signaling, mitotic cell cycle, and calcium signaling (Supplementary Fig. 12d), and showed overlaps with previously reported zygotic transcriptomes of different rice varieties (Supplementary Fig. 13a)[9,20,36]. These genes displayed higher transcription levels in zygote than egg in the different rice varieties and could be clustered based on their expression in egg or sperm cells (Supplementary Fig. 13b). Many were previously reported to associate with ZGA, including *WUSCHEL-related homeobox 5* (*WOX5*), *MINICHROMOSOME MAINTENANCE 6* (*MCM6*), *MCM7/10*, *CYCB2;2*, *Kip-related proteins 1* (*KRP1*), *Rapid alkalinization factor 3* (*RALF3*), and *Anaphase-Promoting Complex 10* (*APC10*)[9,36–39]. In addition, DNA replication such as *POLA4*, *POLD1/4*, *OsRPA1/3* (*Replication protein A*) and 17 histone encoding genes were found in the rice hybrid zygotes (Supplementary Fig. 13b; Supplementary Data 2).

To study the parental contribution to the zygotic gene expression, we analyzed parental SNP reads (2.66 to 6 ×10^6) from the reciprocal hybrid zygote transcriptomes and found that most of the reads were of maternal origin and about 1.5–4.1% of the reads were of paternal origin (Supplementary Fig. 14a). This was consistent with previous results that in rice ZGA occurs in the zygote, with unequal parental contribution where most genes are expressed primarily from the maternal genome[9]. However, egg-produced mRNAs might persist in the early zygote, as observed in Arabidopsis[6,15]. From the SNP reads, we identified 6245 expressed SNP genes (2221 maternal biased, 219 paternal biased) in the MZ zygote and 7116 expressed SNP genes (1666 maternal biased, 262 paternal biased) in the ZM zygote (Supplementary Fig. 14a). Among the SNP genes, 3765 overlapped in the reciprocal hybrids (Supplementary Fig. 14a), of which 1063 were maternal, 28 genes were paternal (Fig. 5a). A number of genes were parental sequence-specific genes. The other genes are mostly enriched in maternal reads in either ZM or MZ zygote, as shown by the density plots (Fig. 5a). The analysis indicated that gene imprinting occurred in the rice zygote. Among the 28 paternal specifically expressed genes (PEGs) in the zygote, only one was found as endosperm-expressed PEGs in rice[40], indicating a different gene imprinting program between zygote and endosperm in rice. Most of the 28 zygotic PEGs were already highly expressed in the sperm (Fig. 5b). Several genes such as *GAMETE EXPRESSED PROTEIN1* (*GEX1*), RALF-like secreted peptide RALF3, and *Arabinogalactan protein 7* (*AGP7*) were shown to function in male gametophyte development and during early embryogenesis[13,41–43]. Recent results showed that *gex1* mutants condition both maternal and paternal effects in early embryogenesis[13], providing genetic evidence that paternal *GEX1* transcripts have a function in early embryos. Nearly all (26/28) of the PEGs showed a low expression in the egg cells (Fig. 5b), nine of which showed hypo DNA methylation in the sperm cells or at the paternal alleles in the zygotes (Fig. 5c), suggesting that PEG might have escaped the zygotic remodeling, as observed in mammals[44]. The maternal alleles of the PEGs could be repressed by other chromatin signatures, such as PRC2-H3K27me3 in Arabidopsis[45]. Most of the 1063 maternal-specifically expressed genes (MEGs) showed expression in the egg cells (Supplementary Fig. 14b), and displayed lower mCHH in egg than in sperm in the upstream region (Supplementary Fig. 14c, d). In the zygotes, the paternal alleles of the MEGs also showed higher mCHH than the maternal alleles (Supplementary Fig. 14e), suggesting that mCHH may be involved in the repression of paternal alleles that likely had also escaped the remodeling process in the zygote.

Analysis of parental allelic-specific reads from the hybrid GE transcriptomes revealed similar numbers of genes with maternal and paternal allelic-specific expression in GE (Supplementary Fig. 15a, b), indicating an increased paternal contribution to gene expression in GE, as observation in Arabidopsis[12,13,15], which was consistent with the reestablishment of the parental allelic-specific DNA methylome in GE. It is shown that increased paternal allele contributions from embryo genes by the globular stage have functional significance in Arabidopsis embryogenesis[13,15]. Analysis of monoallelic gene expression in the reciprocal hybrid GEs identified 102 PEGs and 350 MEGs (Supplementary Fig. 15c), suggesting that gene imprinting persisted till the

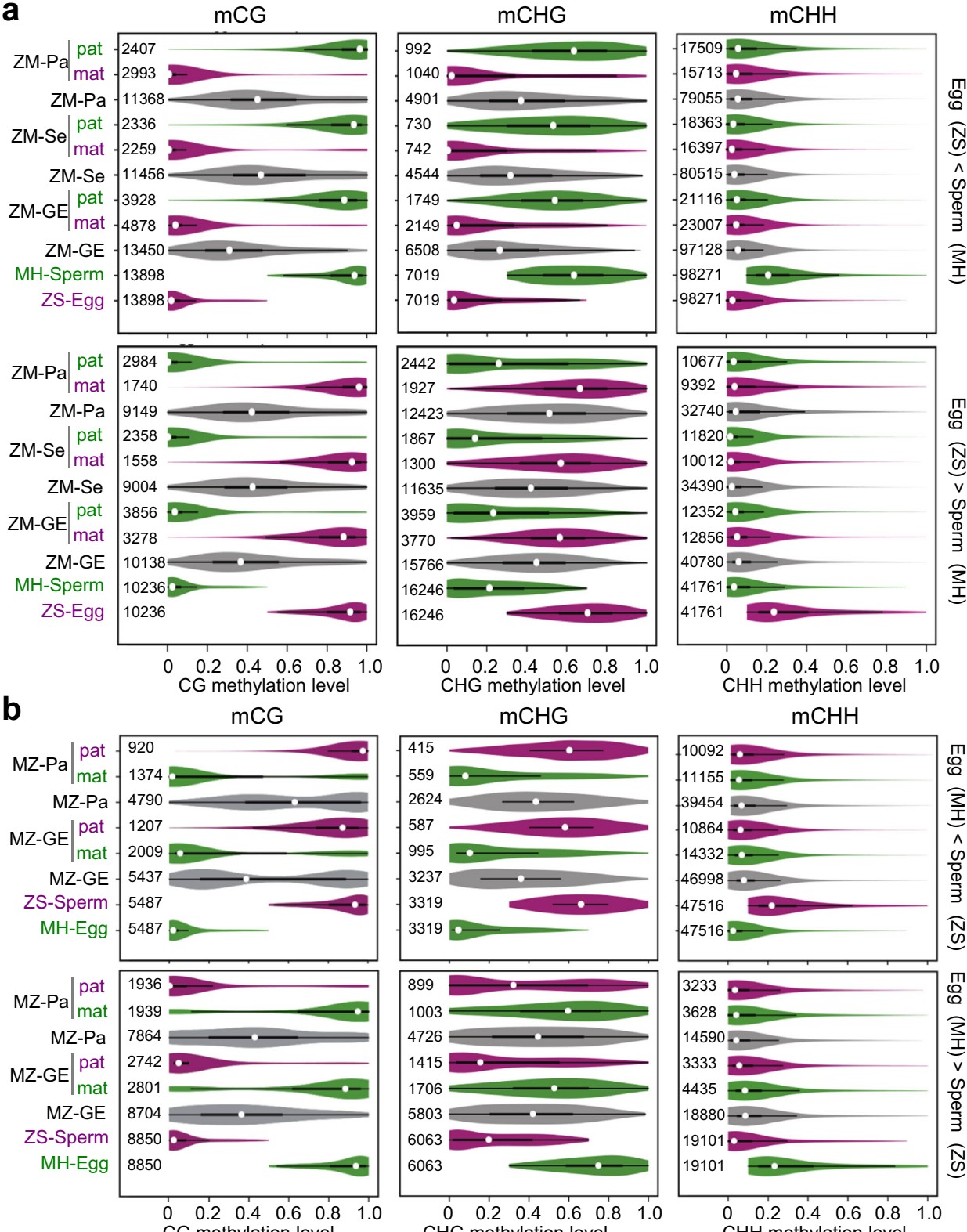

**Fig. 3 | Parental allele-specific methylation levels in hybrid globular embryo, seedling and panicle.** Violin-plots showing the paternal (pat) and maternal (mat) methylation levels of the DMRs in ZM (**a**) and/or MZ (**b**) globular embryos (GE), panicle (Pa) or seedling (Se) in comparison with the respective egg or sperm levels of the parental lines ZS97 (ZS) and MH63 (MH). Figures are the DMR numbers used for the analysis. The horizontal line within the box represents the median, box limits represent the interquartile range (IQR), and whiskers represent 1.5 × IQR. Source data are provided as a Source Data file.

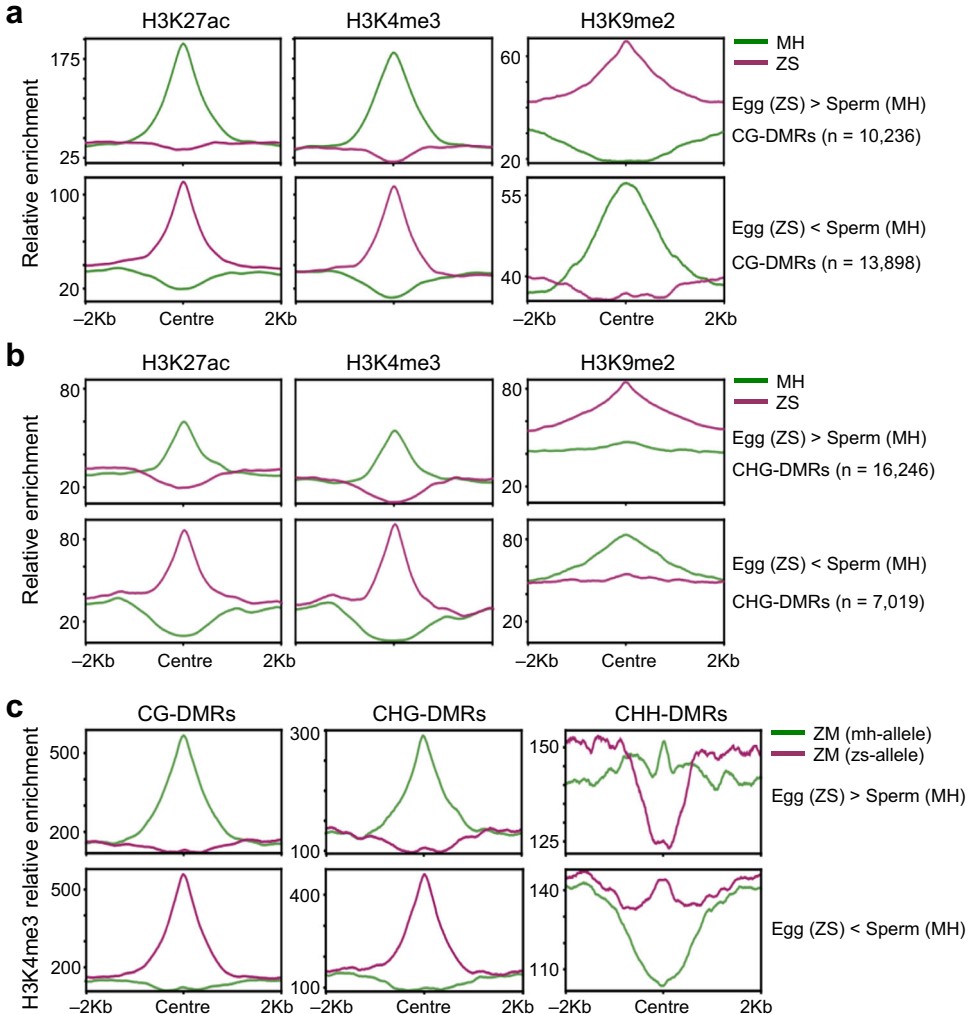

**Fig. 4 | Histone modifications of the egg-sperm DMRs in the parental lines.** H3K27ac, H3K4me3 and H3K9me2 levels of the CG (**a**) and CHG (**b**) DMRs between ZS97 egg and MH63 sperm in MH63 and ZS97 seedlings. Upper panel, hyper-DMRs (ZS97 egg > MH63 sperm); lower panel, hypo-DMRs (ZS97 egg <MH63 sperm).

**c** allelic-specific H3K4me3 of the CG, CHG, and CHH DMRs between ZS97 egg and MH63 sperm in ZM seedling. Upper panel, hyper-DMRs (ZS97 egg > MH63 sperm); lower panel, hypo-DMRs (ZS97 egg <MH63 sperm).

globular embryo stage, which was, however, not detected in the rice mature embryos[46]. The GE imprinted genes were different from those detected in the zygote, except 10% of the zygotic MEGs were remained in GE. Interesting, 36 zygotic MEGs became PEG in GE (Supplementary Fig. 15d).

## Discussion

Epigenetic reprograming is essential for gametogenesis and zygotic development. Sperm cell chromatin is highly condensed but becomes loose after fusion with the egg nucleus allowing transcription of the paternal genome to initiate in the zygote. The predominant paternal DNA methylation remodeling in the zygote may be part of the process. The finding that DNA methylation at a specific set of loci was enhanced in both inbred and hybrid zygotes relative to the gametes, suggests that the parental methylation remodeling is non-stochastic. The finding that maternal methylation-based remodeling of paternal alleles in the rice zygote and in 2-cell embryos corroborates the model that maternal epigenetic pathways control paternal contributions to early embryogenesis in Arabidopsis[12], but contrasts with the findings in zebrafish that after fertilization the maternal genome is reprogrammed to match the paternal methylation pattern that is inherited during early embryogenesis[47,48].

Although the mechanistic details are unclear, maternal epigenetic information and/or regulators inherited from the egg cell may be involved in the process. This is supported by the observations in Arabidopsis that paternal alleles are initially downregulated by the maternal histone H3K9me2 methyltransferase KYP and DNA methyltransferases CMT3 and DRM2[12]. Genes of these enzymes as well as other methylation regulators were found to be expressed at high levels in the rice egg and zygote[49] (Supplementary Fig. 15e, f). The remodeling of paternal allelic DNA methylation to match the maternal allele levels shown in this work likely associates with the zygotic transition to which parental genomes unequally contribute, with most genes expressed primarily from the maternal genome[9,15] (Supplementary Fig. 14a; Fig. 5a). The observations that DNA methylation of the paternal genome was reprogrammed to reach levels similar to the maternal genome, and yet most genes showed predominant maternal expression, suggest that the mechanism by which this maternal-allele preferential expression occurs at an earlier time when the zygote still maintains parental asymmetry in DNA methylation and that the paternal methylation remodeling may contribute to paternal alleles expression in later stages of zygote development.

The reestablishment of paternal allelic-specific methylation observed in rice globular embryos is reminiscent of the re-methylation

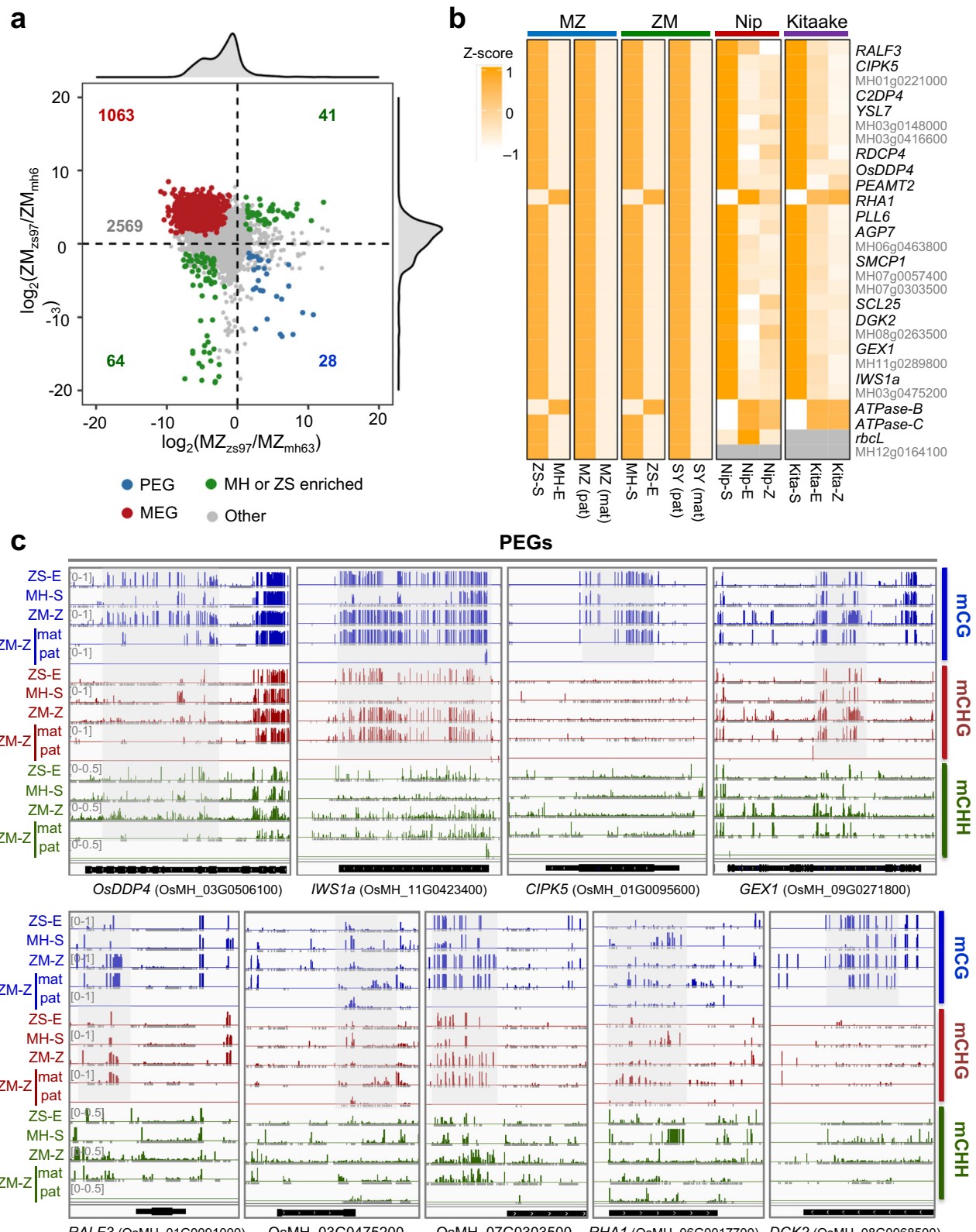

**Fig. 5 | DNA methylation of paternal specifically expressed genes (PEGs) in the hybrid zygotes. a** Identification of paternal specifically expressed genes (PEGs) and maternal specifically expressed genes (MEGs) from the reciprocal hybrid (ZM and MZ) zygotes. Genes in green are parental sequence-specific genes. The other genes are labeled in gray, most of which are enriched in maternal reads shown by the density plots along the x- and y-axis. **b** Expression levels of the PEGs identified from the reciprocal hybrid zygotes in sperm, egg and zygote (with maternal and paternal alleles separated). Genes with functional annotation are in black. **c** Genome browser screenshots of DNA methylation of 9 PEGs in ZM zygote (ZM-Z), ZS97 egg (ZS-E), MH63 sperm (MH-S), paternal-allele in ZM zygote (pat), and maternal-allele in ZM zygote (mat). Source data are provided as a Source Data file.

process in post-implantation embryos in mammals[50], and suggests existence of a memory for parental allelic-specific methylation. Possibly, interplay between parental allelic-specific DNA methylation and histone modifications, which may depend on the associated DNA sequences (*in cis*), may elicit a chromatin memory that facilitates the reestablishment and/or maintenance of parental allelic or sequence-specific epigenetic signatures in the next generation. This, together with the partial DNA methylation remodeling in the gametes and zygote which may spare not only imprinted genes but also other loci, would facilitate transgenerational inheritance of inherent and acquired epigenetic information in plants. Elucidation of mechanisms underlying the setup of parental allelic or sequence-specific methylation memory during plant embryogenesis and its maintenance during development would lead to new strategies for crop improvement.

## Methods

### Rice sperm, egg cell, zygote and GEs isolation

Rice (*Oryza sativa* spp.) varieties Zhenshan 97 (ZS97, *indica/xian*), Minghui 63 (MH63, *indica/xian*), and Zhonghua 11 (ZH11, *japonica/geng*) were used in this study. The three inbred lines were grown in paddy field under normal agricultural conditions in Wuhan, China. To collect hybrid and/or isogenic zygotes, the female lines were hand emasculated and pollinated with the indicated male lines' pollen, and the hybrid zygotes were isolated at 6.5 HAP (for unicellular zygote) and 12 HAP (for two-cell stage zygote), the GEs were isolated at 72 HAP. Egg cell and zygote were isolated from ovaries of rice as previously reported[20,23]. Briefly, ovaries of unpollinated and pollinated florets were manually dissected under dissection microscope. Then the dissociated ovule was transferred into 0.53 M mannitol solution (Sigma) and broken to release egg cell or zygote. The isolated cells were stained with fluorescein diacetate (Invitrogen, Cat. # F1303) and collected by a micromanipulator system (Eppendorf, TransferMan 4r). Twenty-five egg or zygote cells were pooled for each replicate for BS-seq or RNA-seq libraries, each cell-type or each genotype with three biological replicates. Sperm cells were collected as previously reported method[51,52] with minor modifications. Briefly, about 30 anthers were collected from mature florets before anthesis in a plastic dishes with 3 mL of 12% sucrose, then broken with forceps to release pollen. Sperm cells were released by gentle shaking for 30 min and filtered through 20 μm and then 10 μm nylon bolting clothes. Subsequent steps were performed as described[51,52].

### RNA-seq and BS-seq library construction and sequencing

For RNA-seq library construction, mRNAs were extracted from the collected rice gamete and zygote cells, then reverse transcribed and amplified by using a Single Cell Full Length mRNA Amplification Kit (Vazyme, Cat.# N712) according to manufacturer's instruction. cDNAs were purified with VAHTS DNA Clean Beads (Vazyme, Cat.# N411) and fragmented into 200 ~ 500 bp lengths, then used for PCR amplification, adapter/index ligation, and DNA purification with a TruePrep® DNA Library Prep Kit V2 for Illumina (Vazyme, Cat.# TD502). BS-seq libraries were constructed using a previously reported protocol[24] with modified primer adapter 2 oligos and iPCRtag primers[20]. RNA-seq and BS-seq libraries were sequenced by an Illumina NovaSeq 6000 platform (Annoroad Gene Technology, China) with the PE150 (paired-end 150 nucleotides) method.

### RNA-seq data analysis

RNA-seq raw reads were filtered by fastp[53] (v.0.20.1) to remove low-quality reads and adapter. Clean reads were aligned to the MH63 reference genome (MH63RS3, Rice Information GateWay [RIGW], http://rice.hzau.edu.cn/rice_rs3/) by HISAT2[54] (v.2.2.1). To improve alignment of ZS97 RNA-seq data, a pseudogenome was constructed by using the MH63 genome as backbone and replacing the SNPs (between MH63 and ZS97) with ZS97 genotype to map ZS97 sequencing reads. The unique mapping reads were retained for further analysis. StringTie[55] (v.2.1.4) was

used for transcripts assembly and gene quantitation. DESeq2[56] package was used for gene differential expression analysis. Genes with TPM (transcripts per million) ≥ 1 (at least in one sample in the comparisons) and with |log2 (fold change)| ≥ 2 and adjusted $P < 0.01$ were considered as differentially expressed genes (DEGs).

For allele-specific expression (ASE) analysis of the hybrids, the SNPs between MH63 and ZS97 were masked with N by using SNPsplit[57] (v.0.3.4). Clean reads were aligned on the N-masked MH63 genome by HISAT2 (v.2.2.1) and the unique mapping reads were retained. The parental allele-specific reads were separated from the hybrids data by using SNPsplit program. The separated reads were normalized for allelic-specific expression level calculation. Allele-specific expression genes were identified with the cut-offs |log2 (fold change)| > 1 and adjusted $P < 0.01$ between two parental alleles by DESeq2 package.

### BS-seq data analysis

BS-seq low-quality reads were filtered out from the raw data by Trim_Galore (v.0.6.6; http://www.bioinformatics.babraham.ac.uk/projects/trim_galore/). Clean reads were aligned on the MH63 genome by Bismark (v.0.23.1)[58] using default parameters. ZS97 BS-seq reads were aligned on the SNP-N-masked MH63 genome. Unique mapping reads were retained for further analysis. PCR duplications were removed by command of deduplicate_bismark and DNA methylation sites were extracted by command of bismark_methylation_extractor from Bismark software (v.0.23.1). Individual cytosines with more than three reads were retained for DNA methylation level calculation.

For allele-specific methylation analysis, the SNPs between MH63 and ZS97 were masked with N by SNPsplit (v.0.3.4). The cleaned high-quality reads were mapped to the N-masked MH63 genome by Bismark. After removing duplications, the allele-specific reads were separated from the hybrids by the SNPsplit. Individual cytosines that were covered by at least three allele-specific reads were considered for allele-specific methylation level calculation.

To identify differential methylated regions (DMRs), the whole genome was divided into 50-bp bins. Bins that contained at least five cytosines each and every cytosine with at least a three-fold coverage were retained. Bins with methylation differences greater than 0.5, 0.3, and 0.1 respectively at CG, CHG, and CHH contexts with false discovery rate (FDR) < 0.05 between comparisons were considered as DMRs. The FDR was generated from an adjusted $P$-value (Fisher's exact test) using the Benjamini-Hochberg method.

Density plots were generated by comparing the average cytosine methylation levels within 50 bp bins between two samples. Only the bins contained at least 20 informative sequenced cytosines (i. e., the sum of the sequence depth of each cytosine multiplied by the number of cytosines within 50 bp bins in the CG, CHG, or CHH context) in both samples and 0.5 CG, 0.3 CHG, or 0.1 CHH methylation ratios in either sample were retained as previously described[20,21]. The frequency distribution of fractional methylation differences between comparisons was shown by density plots. Genomic distribution of the CHH DMRs between hybrid zygotes and gametes were visualized by circos plots using TBtools[59].

### Chromatin immunoprecipitation-sequencing (ChIP-seq) and data analysis

Chromatin immunoprecipitated experiments were conducted as previously described[60]. Briefly, about 2 g of rice seedling leaves were crosslinked by 1% (v/v) formaldehyde for 30 min and used for chromatin extraction. Chromatin was fragmented to around 200 bp by sonication using a Bioruptor Plus System (Diagenode), and then incubated with antibody-conjugated beads (anti-H3K4me3, Abcam Cat.# ab8580) overnight. After washing three times, immunoprecipitated chromatin was de-crosslinked and DNA was purified, non-precipitated chromatin was used as input. DNA isolated from chromatin immunoprecipitation was used for sequencing libraries construction according to the protocol

of Illumina TruSeq ChIP Sample Prep Set A and sequenced on Illumina HiSeq2500 platform.

Fastp (v.0.20.1) was used for remove low-quality reads and adapter from the ChIP-seq raw data. Clean reads were mapped to the MH63RS3 genome by Bowtie2 (v.2.2.8). ZS97 sequencing reads were mapped to the SNP-N-masked MH63 genome. Duplications were removed using Picard (v.2.1.1). The bigwig files were generated by using a command of bamCoverage from deepTools (v.3.3.0). The ChIP-seq data of the hybrids were aligned to the SNP-N-masked MH63 genome by Bowtie2 (v.2.2.8). SNPsplit (v.0.3.4) software was used to separate the parental allele-specific modification reads.

### Reporting summary
Further information on research design is available in the Nature Portfolio Reporting Summary linked to this article.

## Data availability
The BS-seq, RNA-seq, and ChIP-seq data generated in this study are deposited into the NCBI Sequence Read Archive (BioProject ID: PRJNA957147). BS-seq and ChIP-seq data of 93–11 and Nipponbare were downloaded from the NCBI (BioProject ID: PRJNA514100; PRJNA434178). BS-seq data of Nipponbare and Kitaake reproduction cells were respectively downloaded from the DNA Data Bank of Japan (ID: DRA007969) and NCBI (SRP119200). ChIP-seq data (H3K9me2, H3K4me3, H3K27ac) of MH63 and ZS97 leaf were downloaded from the NCBI GEO under accession number GSE142570. Source data are provided with this paper.

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

## Acknowledgements

We would like to thank Mr. Qinghua Zhang and Dr. Xianghua Li for assistance. We thank Tong Hu and Xin Ming for help with cell collect. Computation resources were provided by the high-throughput computing platform of the National Key Laboratory of Crop Genetic Improvement at Huazhong Agricultural University and supported by Hao Liu. The work was supported by the National Natural Science Foundation of China (31821005), the Fundamental Research Funds for the Central Universities (2662015PY228), and the French Agence Nationale de la Recherche (ANR-219CE20-0012-01).

## Author contributions

X.M. collected cell samples and performed libraries construction; Q.L. participated in libraries construction, did bioinformatics analysis and data mining, X.L., X.Z. and S.Z. participated in experimental work; L.X. and Y.Z. participated in the project supervision and management; D.X.Z. conceived and supervised the project, wrote and revised the paper with inputs from Q.L. and X.M.

## Competing interests

The authors declare no competing interests.
