## [Peer Review File · Nature Communications]

Paternal DNA methylation is remodeled to maternal levels in rice zygoteReviewer #1 (Remarks to the Author):

The manuscript by Liu et al reports the results of epigenomic and transcriptomic experiments on rice eggs, sperm, zygotes, and hybrid zygotes and globular embryos resulting from crosses of the MH63 and ZS97 varieties. Using bisulfite sequencing of eggs, sperm, zygotes, and hybrid zygotes, the authors provide evidence that the methylome of the paternal genome is remodeled in the zygote to a methylation status similar to the maternal genome. Interestingly, the authors provide evidence that, by the globular stage, the methylation patterns of the maternal and paternal genomes return to their overall patterns at the time of fertilization, ie. to the methylation patterns in the gametes. The authors then correlate differentially methylated regions between maternal and paternal genomes with histone marks found in seedling tissues, providing a nice association between parent-of-origin methylation status in hybrid zygotes and histone marks in the respective parental lines. Finally, the authors produce transcriptomes from eggs and sperm of the MH63 and ZS97 varieties, as well as hybrid zygotes and globular embryos from crosses in both directions. The authors compare the transcript levels from the maternal and paternal alleles with epigenetic marks in the egg, sperm and zygote, finding correlations between parent-of-origin gene expression and different contexts of cytosine methylation.

This manuscript describes novel experiments conducted on epigenetic regulation of parent-of-origin gene expression in plant embryogenesis. The field of early embryogenesis in plants has been waiting for this combination of epigenomic and transcriptomic data of gametes and hybrid zygotes for a long time. This manuscript is absolutely appropriate for publication in Nature Communications, if not Nature. The manuscript would benefit from some light editing for language style. Below I list comments and suggestions for improving the manuscript.

Stewart Gillmor
Langebio, CINVESTAV, Mexico

Figure 1C: why is there a clear bi-modal distribution (some genes lose methylation and some gain methylation) but only in the CHH context? Are these genes enriched for some GO term or in a particular region of the genome like near centromeres or transposons?

Extended Data Figure 10a: Reads are shown only as % maternal or % paternal, without regard to how many genes these reads map to. The authors should also show how many genes were called as having statistically significant maternal or paternal bias.

Figure 5a The authors should show how many genes do not show maternal or paternal bias (i.e. how many genes are represented by a grey point).

line 18 'DNA methylation'

line 22 'reprogramming'

line 31-32 for a more in-depth review on parent-of-origin contributions to early plant embryogenesis which includes a discussion of genetic as well as transcriptomic studies, the authors might consider also citing Armenta-Medina and Gillmor <https://doi.org/10.1016/bs.ctdb.2018.11.008>

lines 34-35 when discussing the contrasting results in Arabidopsis on parent-of-origin studies of hybrid zygote and early embryo transcriptomes, the authors should cite Alaniz-Fabián et al 2022 <https://doi.org/10.1242/dev.201025>, who reanalyzed the Col/Ler zygote transcriptomes originally described in Zhao et al (2019) Dev Cell and Zhao et al (2020) Nature Plants. This reanalysis showed that on a gene-by-gene bases, Col/Ler zygotes do not show equal parental transcriptome contributions; thousands of genes in Col/Ler zygotes are represented by transcripts from either the maternal or paternal allele, but not both. Thus, previous conclusions on parent-of-origin transcript contributions are oversimplified, and the evidence for equal transcript contributions in Col/Ler zygotes is not as strong as originally presented.

Consistent with Liu et al's finding that maternal transcripts dominate in rice zygotes, previous work

in Arabidopsis by Alaniz-Fabián et al (2022) Development as well as Del Toro De León et al (2014) Nature presented genetic evidence that maternal alleles of most EMB genes make a more important contribution functional to early embryogenesis than paternal alleles, and that hybridization itself can affect parental genome contributions to early embryogenesis. Adding a sentence about this genetic evidence for parental genome contributions in Arabidopsis would provide a fuller picture of work on parent-of-origin regulation of early embryogenesis in plants, and would put the experiments presented by Liu et al in a more complete context.

lines 61-63 'DNA methylomes data were obtained from 25 eggs or zygotes and 150 sperm cells, two biological replicates were performed with a sequencing depth of about 24.7 - 75.4 × genome coverage (Supplementary Table 1).' Question: Is this 25x Coverage from 25 eggs? After excluding duplicates?

lines 64-64 'sperm methylome more distal from that of egg or zygote' An inspection of Extended Data Figure 1a does not support this statement. Only for mCHH of the ZS97 variety is sperm methylation separated from egg and zygote, for other methylation types and the other variety they are not separated. The authors should remove this statement.

Lines 66-67 'Boxplots indicated that sperm cells showed globally higher CG methylation (mCG) but lower CHG methylation (mCHG) than egg cells' An inspection of Extended Data Figure 2a does not fully support this statement. Only MH63 apparently has higher CG for sperm than egg. Also, are the differences shown by the box plots statistically significant, even when the median looks different? The authors should do statistical analysis to determine if the results in the box plots in Extended Data Figure 2 and in Figure 1 really are different.

Lines 70-71 'In the zygote mCG and mCHH levels were lower than in the sperm, while the mCHG was at the intermediate levels of the egg and sperm cells' Same as my previous comment- are the differences in box plots statistically significant?

line 85 I find the notation that the authors use to describe the different hybrids confusing. The MH63 x ZS97 hybrid is notated as MZ; this makes sense. But the ZS97 x MH63 hybrid is notated as SY63. I understand that this is a standard hybrid in rice and that is its name, but perhaps for the purposes of simplicity in the manuscript the authors could refer to the SY63 hybrid as ZM?

Lines 86-88 'In the hybrid zygotes, the methylation levels appeared higher than in the male and female gametes, particularly at CG and CHG sites (Fig. 1)' As I mentioned above, the authors should conduct statistical tests to determine if the data represented in the box plots in Figure 1 are statistically significantly different.

Lines 117-118 'indicating that the maternal-controlled remodeling of paternal allele-specific methylation in the zygote persisted till at least the 2-cell embryo stage.' The authors do not formally show that the remodeling that occurs is maternally controlled. All they show is that the remodeling makes the paternal genome similar to the maternal genome. They should soften this statement.

Lines 191-193 'Several genes such as plasma membrane protein gene GEX1, RALF-like secreted peptide RALF3, and Arabinogalactan protein 7 (AGP7) were shown to function in male gametophyte development and during early embryogenesis 39-41'. Alaniz-Fabián (2022) Development showed that gex1 mutants condition both maternal and paternal effects in early embryogenesis. This is genetic evidence that paternal GEX1 transcripts have a function in early embryos, consisted with GEX1 being a PEG. The authors might consider citing Alaniz-Fabián (2022) here.

Lines 207-209 'indicating an increased paternal contribution to gene expression in GE, as observation in Arabidopsis 11, which was consistent with the reestablishment of the parental allelic-specific DNA methylome in GE'. In addition to citing the transcriptomic data of Autran et al that indicates equal maternal and paternal transcriptome contributions at the globular stage, the authors might consider also citing Del Toro De León (2014) Nature and Alaniz-Fabián (2022) Development, which provide functional evidence for increased paternal allele contributions from

EMB genes by the globular stage of embryogenesis in Arabidopsis.

Reviewer #2 (Remarks to the Author):

The manuscript entitled "Maternal-based paternal DNA methylation remodeling in rice zygote" aims to decipher how DNA methylation (DNAm) is remodeled after fertilization. The authors provide a very comprehensive experimental dataset of DNAm on egg, sperm, zygote and globular embryos. Results described here provide evidence of how DNAm in the zygote changes after fertilization to match a pattern that closely resembles the maternal genome. The data also shows how hybridization has immediate effects on DNAm in a subset of loci in zygotes and globular embryos.

Bellow you can find a summary of my comments by sections of the present manuscript.

"Predominant remodeling of the male methylome in the rice zygote upon fertilization"

This first section of the manuscript describes global comparisons of DNA methylation in S, E and Z. However, the title and the conclusions drawn refer to a remodeling of the male methylome. This is indeed suggested by the data, but convincingly demonstrated later in the manuscript with parent-specific DNA methylation. I suggest changing the title of this section to more accurately reflect the findings described in it.

Line 72: "Density plots revealed higher methylation variations between zygote and sperm than between zygote and egg (Extended Data Fig. 2b)." Results and interpretations on this part feel disjointed. For clarity, please rephrase by explaining the quantitative differences observed in the density plots that support the statements.

"A number of given loci tend to be remodeled in the zygote"

In this section, the authors highlight that in zygotes, "DNA methylation at a number of specific loci tends to be enhanced upon fertilization," which is particularly true in hybrid zygotes. What are these loci? Do they have anything in common? And how this reflects on gene expression?

Line 88: "(Fig. 1)", what part of the panel is this referring to?

"Parental allele-specific methylation was restored during embryogenesis and stably maintained in the hybrids"

Here the authors nicely show that although the DMRs between egg and sperm are closer to the maternal genome in hybrids, they recover the parent-specific profile by the globular stage. I wonder if this is consistent with the activity of the RdDM pathway during embryogenesis? The authors have RNAseq data that can help answer this question.

Figure 3. Label x axis.

"Parental methylation difference was associated with distinct histone modifications"

Figure 4. I suggest including at least in the extended data a representation of the data using box plots, which are more quantitative and less sensitive to bias from individual loci.

"Parental DNA methylation remodeling mirrors parental contribution to zygotic gene expression"

Whether DNA methylation reprogramming after fertilization has an effect on ZGA is still unclear and not discussed by the authors. The fact that DNA methylation of the paternal genome is reprogrammed to reach levels similar to the maternal genome, and yet most genes show predominant maternal expression, suggests that the mechanism by which this maternal-allele preferential expression occurs is independent from DNA methylation, or that it occurs at an earlier time when the zygote still maintains parental asymmetry in DNA methylation.

Line 179: Not clear. What do they mean with "most of the reads were maternal allelic-specific and about 1.5%~4.1% of the reads was paternal allelic-specific"? I find hard to reconcile this statement with figure 5a where most genes are biallelic (with a level of maternal bias expression) while a small subset of genes shows either maternal (1063) or paternal (28) specific expression.

The authors here refer to Extended Data Figure 10a, which shows a Venn diagram showing, I believe, the parental reads for MEGs in zygotes or for the entire transcriptome? Please rephrase this paragraph for clarity. In the Extended Data Figure 10a, include the total number in Venn diagrams.

Line 18: DNA

Reviewer #3 (Remarks to the Author):

In the manuscript "Maternal-based paternal DNA methylation remodeling in rice zygote", authors Liu et al investigate changes in DNA methylation that occur in early rice zygotes. They observe that in regions of the genome with differential methylation between egg and sperm, zygotes show more similarity to the egg cell methylation levels resulting from changes in paternal methylation. As observed in the literature previously, DNA methylation patterns are consistent across development, with a major exception being the sperm cell, so remodeling of paternal methylation is expected.

The authors hypothesized that the maternal genome is acting to "control" the remodeling of the paternal genome, however none of the experiments or analyses demonstrated active involvement of the maternal genome. This is important because although the zygote is more correlated with the maternal than paternal epigenome, so is the DNA methylation profile of all vegetative tissue in the plant. Sperm cells have active remodeling of methylation during development, so the change in paternal methylation is necessary to regain the methylation pattern typical of vegetative cells. In order to make the claim that "paternal DNA methylation remodeling" is "maternal-based" as stated in the title and throughout the manuscript, the authors would need to somehow distinguish between paternal methylation reverting to vegetative levels vs. changing to maternal levels. While some cases of trans-acting epi-alleles exist in plants (i.e. paramutation), DNA methylation is also inherited trans-generationally and many alleles maintain methylation patterns consistent with the genetic origin of that sequence, as demonstrated in this paper as well.

In order to make the claim that DNA methylation remodeling is maternal-based, the authors need to exclude the more likely possibility that DNA methylation is simply being re-established to vegetative levels based on cis-regulatory information coded in the DNA. A cis-regulatory mechanism of DNA methylation reestablishment would also explain why the maternal shift occurs for CG and CHG but not CHH methylation which depends on trans-acting factors. The genotype-dependent methylation patterns are demonstrated in Figure 3, however, the authors label this as parental effects instead of genotype effects. Excluding re-establishment of vegetative methylation could potentially be performed with additional analyses of existing data, however it may require additional experiments to demonstrate persuasively.

In addition to the major problem of the data not matching the main point of the paper, there are several smaller cases of the text of the manuscript not matching or oversimplifying the results shown on figures. I have listed these examples below.

Line 64 – PCA shows separation by genotype before cell type. Please rewrite to reflect observations.

Line 66 – Boxplots show slightly lower CG for sperm in one genotype but higher CG in the other genotype. CHG is very similar.

Line 78 – Only some of the bars show that "more than half of DMRs concerned non-TE regions.

Line 87 – Methylation was higher for genes but not TEs.

Line 200 – The figures do not look "overall higher" in all regions, but does look marginally higher in upstream and downstream regions. Error bars or other stats on the metaplots would help clarify if the signal here is significant.

Line 206 – What is meant by “comparable numbers”? It looks like fewer MEGs and PEGs than zygotes.

Finally, the authors analyze parent-of-origin biased gene expression and conclude that expression patterns are dynamic and shift between MEGs, PEGs, and non-imprint. However, these calls are only described as discreet data. The problem with this is that imprinting calls are typically based on a small number of allele-specific reads, and a change from, for example, MEG to not imprinted can reflect a real change in allelic expression, a loss of expression of the expressed allele, or most often, it can reflect a subtle change that is no longer called significant without necessarily reflecting the biology. In order to conclude that imprinting is dynamic in these early stages, the authors need to demonstrate that the change in patterns is real and not an artifact of small numbers of reads.

Point-by-point response to the reviewers

Reviewer #1 (Remarks to the Author):

The manuscript by Liu et al reports the results of epigenomic and transcriptomic experiments on rice eggs, sperm, zygotes, and hybrid zygotes and globular embryos resulting from crosses of the MH63 and ZS97 varieties. Using bisulfite sequencing of eggs, sperm, zygotes, and hybrid zygotes, the authors provide evidence that the methylome of the paternal genome is remodeled in the zygote to a methylation status similar to the maternal genome. Interestingly, the authors provide evidence that, by the globular stage, the methylation patterns of the maternal and paternal genomes return to their overall patterns at the time of fertilization, ie. to the methylation patterns in the gametes. The authors then correlate differentially methylated regions between maternal and paternal genomes with histone marks found in seedling tissues, providing a nice association between parent-of-origin methylation status in hybrid zygotes and histone marks in the respective parental lines. Finally, the authors produce transcriptomes from eggs and sperm of the MH63 and ZS97 varieties, as well as hybrid zygotes and globular embryos from crosses in both directions. The authors compare the transcript levels from the maternal and paternal alleles with epigenetic marks in the egg, sperm and zygote, finding correlations between parent-of-origin gene expression and different contexts of cytosine methylation.

This manuscript describes novel experiments conducted on epigenetic regulation of parent-of-origin gene expression in plant embryogenesis. The field of early embryogenesis in plants has been waiting for this combination of epigenomic and transcriptomic data of gametes and hybrid zygotes for a long time. This manuscript is absolutely appropriate for publication in Nature Communications, if not Nature. The manuscript would benefit from some light editing for language style. Below I list comments and suggestions for improving the manuscript.

Stewart Gillmor

Langebio, CINVESTAV, Mexico

Figure 1C: why is there a clear bi-modal distribution (some genes lose methylation and some gain methylation) but only in the CHH context? Are these genes enriched for some GO term or in a particular region of the genome like near centromeres or transposons?

Response: Thank you for the comments. As suggested, we analyzed the genomic distribution of the CHH DMRs in zygotes vs the gametes and found that the hyper-DMRs were mainly enriched in genic regions, and the hypo-DMRs enriched in TE regions, which is consistent with the observations that sperm and egg show high levels of mCHH in long TEs. The GO enrichment indicates that the concerned genes are mainly enriched in RNA silencing, defense and developmental pathways.

The analysis is presented in **Supplementary Fig. 4** and commented in the results section:

“Density plots revealed a clear bimodal distribution pattern of CHH DMR between zygote and sperm (Z – S) or between zygote and egg (Z – E) (**Figure 1c**), indicating a fraction of loci showed clearly increased (hyper) or decreased (hypo) methylation at CHH sites in the zygote genome. Further analysis indicated that the hyper methylated CHH sites were enriched in genic regions whereas the hypo-methylated sites were mainly located in TE regions (**Supplementary Fig. 4a**). Genes with the CHH DMRs were mainly enriched in RNA silencing, defense and developmental pathways” (**Supplementary Fig. 4b**).

Extended Data Figure 10a: Reads are shown only as % maternal or % paternal, without regard to how many genes these reads map to. The authors should also show how many genes were called as having statistically significant maternal or paternal

bias.

Response: Thanks for the comments. As suggested, we have added the informative sequencing reads and the numbers of expressed SNP genes (maternal and paternal) in the Supplementary Fig. 14a) in the revised version.

Figure 5a The authors should show how many genes do not show maternal or paternal bias (i.e. how many genes are represented by a grey point).

Response: Thanks for the suggestion. We have reanalyzed the zygotic genes. The gene number (2569) represented by the grey points is added in Fig.5a in the revised version.

line 18 'DNA methylation'

Response: Thanks, we have corrected the typo.

line 22 'reprogramming'

Response: Corrected.

line 31-32 for a more in-depth review on parent-of-origin contributions to early plant embryogenesis which includes a discussion of genetic as well as transcriptomic studies, the authors might consider also citing Armenta-Medina and Gillmor <https://doi.org/10.1016/bs.ctdb.2018.11.008>

Response: Thank you for the point. We have included the sentence “Parent-of-origin contributions to plant early embryogenesis have been studied at genetic and transcriptomic levels¹⁰.” and cited the reference in the revised version.

lines 34-35 when discussing the contrasting results in Arabidopsis on parent-of-origin studies of hybrid zygote and early embryo transcriptomes, the authors should cite

Alaniz-Fabián et al 2022 <https://doi.org/10.1242/dev.201025>, who reanalyzed the Col/Ler zygote transcriptomes originally described in Zhao et al (2019) Dev Cell and Zhao et al (2020) Nature Plants. This reanalysis showed that on a gene-by-gene basis, Col/Ler zygotes do not show equal parental transcriptome contributions; thousands of genes in Col/Ler zygotes are represented by transcripts from either the maternal or paternal allele, but not both. Thus, previous conclusions on parent-of-origin transcript contributions are oversimplified, and the evidence for equal transcript contributions in Col/Ler zygotes is not as strong as originally presented.

Response: Thank you for the comments. We have mentioned the results of reanalysis and cited the references in the revised version.

“However, a reanalysis of the published data^{6,7} showed that, on a gene-by-gene basis, the Arabidopsis hybrid (Col/Ler) zygotes do not show equal parental transcriptome contributions; thousands of genes in hybrid zygotes are represented by transcripts from either the maternal or paternal allele, but not both¹³.”

Consistent with Liu et al's finding that maternal transcripts dominate in rice zygotes, previous work in Arabidopsis by Alaniz-Fabián et al (2022) Development as well as Del Toro De León et al (2014) Nature presented genetic evidence that maternal alleles of most EMB genes make a more important contribution functional to early embryogenesis than paternal alleles, and that hybridization itself can affect parental genome contributions to early embryogenesis. Adding a sentence about this genetic evidence for parental genome contributions in Arabidopsis would provide a fuller picture of work on parent-of-origin regulation of early embryogenesis in plants, and would put the experiments presented by Liu et al in a more complete context.

Response: Thanks for the suggestion. We have added the context in the revised Introduction and cited the related references.

“There is also genetic evidence that maternal alleles of most embryo genes make a more important contribution functional to early embryogenesis than paternal alleles,

and that hybridization itself can affect parental genome contributions to early embryogenesis^{13,15}.”

lines 61-63 'DNA methylomes data were obtained from 25 eggs or zygotes and 150 sperm cells, two biological replicates were performed with a sequencing depth of about 24.7 - 75.4 × genome coverage (Supplementary Table 1).' Question: Is this 25x Coverage from 25 eggs? After excluding duplicates?

Response: Genome coverage was calculated per replicate (25 cells). We have clarified this in the MS.

“DNA methylomes data were obtained from 25 eggs or zygotes and 150 sperm cells, two biological replicates were performed with a sequencing depth of about 24.7 - 75.4 × genome coverage per replicate.”

lines 64-64 'sperm methylome more distal from that of egg or zygote' An inspection of Extended Data Figure 1a does not support this statement. Only for mCHH of the ZS97 variety is sperm methylation separated from egg and zygote, for other methylation types and the other variety they are not separated. The authors should remove this statement.

Response: Thanks for the point. We have rephrased the statement:

“Principal component analysis revealed a high reproducibility of the replicates and a clear difference between the two parental lines”.

Lines 66-67 'Boxplots indicated that sperm cells showed globally higher CG methylation (mCG) but lower CHG methylation (mCHG) than egg cells' An inspection of Extended Data Figure 2a does not fully support this statement. Only MH63 apparently has higher CG for sperm than egg. Also, are the differences shown by the box plots statistically significant, even when the median looks different? The

authors should do statistical analysis to determine if the results in the box plots in Extended Data Figure 2 and in Figure 1 really are different.

Response: Thanks for the comments. We have resented the statement and added the statistical significance of the boxplots in the revised version.

“Boxplots indicated that sperm cells showed globally lower CHG methylation (mCHG) than egg cells in TEs.”

Lines 70-71 'In the zygote mCG and mCHH levels were lower than in the sperm, while the mCHG was at the intermediate levels of the egg and sperm cells' Same as my previous comment- are the differences in box plots statistically significant?

Response: We have added the statistical significance of the boxplots in the revised Figures.

line 85 I find the notation that the authors use to describe the different hybrids confusing. The MH63 x ZS97 hybrid is notated as MZ; this makes sense. But the ZS97 x MH63 hybrid is notated as SY63. I understand that this is a standard hybrid in rice and that is its name, but perhaps for the purposes of simplicity in the manuscript the authors could refer to the SY63 hybrid as ZM?

Response: We have changed SY63 to ZM as advised.

Lines 86-88 'In the hybrid zygotes, the methylation levels appeared higher than in the male and female gametes, particularly at CG and CHG sites (Fig. 1)' As I mentioned above, the authors should conduct statistical tests to determine if the data represented in the box plots in Figure 1 are statistically significantly different.

Response: We have added statistical significance of the boxplots in the revised Figures.

Lines 117-118 'indicating that the maternal-controlled remodeling of paternal

allele-specific methylation in the zygote persisted till at least the 2-cell embryo stage.' The authors do not formally show that the remodeling that occurs is maternally controlled. All they show is that the remodeling makes the paternal genome similar to the maternal genome. They should soften this statement.

Response: We have rewritten the statement in the revised version.

“The data together indicated that the paternal allele-specific methylation is remodeled to the levels similar to the maternal alleles in the zygote, which persists till at least the 2-cell embryo stage.”

Lines 191-193 'Several genes such as plasma membrane protein gene GEX1, RALF-like secreted peptide RALF3, and Arabinogalactan protein 7 (AGP7) were shown to function in male gametophyte development and during early embryogenesis 39-41'. Alaniz-Fabián (2022) Development showed that *gex1* mutants condition both maternal and paternal effects in early embryogenesis. This is genetic evidence that paternal GEX1 transcripts have a function in early embryos, consisted with GEX1 being a PEG. The authors might consider citing Alaniz-Fabián (2022) here.

Response: Thanks for the suggestion. We have discussed the point and cited the reference in the revised version.

“Recent results showed that *gex1* mutants condition both maternal and paternal effects in early embryogenesis¹³, providing genetic evidence that paternal *GEX1* transcripts have a function in early embryos.”

Lines 207-209 'indicating an increased paternal contribution to gene expression in GE, as observation in Arabidopsis 11, which was consistent with the reestablishment of the parental allelic-specific DNA methylome in GE'. In addition to citing the transcriptomic data of Autran et al that indicates equal maternal and paternal transcriptome contributions at the globular stage, the authors might consider also citing Del Toro De León (2014) Nature and Alaniz-Fabián (2022) Development,

which provide functional evidence for increased paternal allele contributions from EMB genes by the globular stage of embryogenesis in Arabidopsis.

Response: Thanks for the suggestion. We have discussed the finding and cited the references in the revised version:

“It is shown that increased paternal allele contributions from embryo genes by the globular stage have functional significance in Arabidopsis embryogenesis^{13,15}.”

Reviewer #2 (Remarks to the Author):

The manuscript entitled "Maternal-based paternal DNA methylation remodeling in rice zygote" aims to decipher how DNA methylation (DNAm) is remodeled after fertilization. The authors provide a very comprehensive experimental dataset of DNAm on egg, sperm, zygote and globular embryos. Results described here provide evidence of how DNAm in the zygote changes after fertilization to match a pattern that closely resembles the maternal genome. The data also shows how hybridization has immediate effects on DNAm in a subset of loci in zygotes and globular embryos.

Bellow you can find a summary of my comments by sections of the present manuscript.

“Predominant remodeling of the male methylome in the rice zygote upon fertilization”

This first section of the manuscript describes global comparisons of DNA methylation in S, E and Z. However, the title and the conclusions drawn refer to a remodeling of the male methylome. This is indeed suggested by the data, but convincingly demonstrated later in the manuscript with parent-specific DNA methylation. I suggest changing the title of this section to more accurately reflect the findings described in it.

Response: Thank you for the comments. As suggested, we have changed the title of this section in the revised version to:

“Remodeling of the rice gamete methylomes in the zygote upon fertilization”

Line 72: “Density plots revealed higher methylation variations between zygote and sperm than between zygote and egg (Extended Data Fig. 2b).” Results and interpretations on this part feel disjointed. For clarity, please rephrase by explaining the quantitative differences observed in the density plots that support the statements.

Response: Thanks for the point. We have rewritten the statements in the revised version:

“Density plots revealed lower mCG in zygote relative to both sperm and egg, and lower mCHH but higher mCHG in zygote versus sperm”

“A number of given loci tend to be remodeled in the zygote”

In this section, the authors highlight that in zygotes, “DNA methylation at a number of specific loci tends to be enhanced upon fertilization,” which is particularly true in hybrid zygotes. What are these loci? Do they have anything in common? And how this reflects on gene expression?

Response: Thank for comments. We have analyzed the overlapping hyper DMRs between hybrid and inbred zygote relative sperm/egg and found many genes are associated with the DMRs. GO enrichment indicates that the genes are of diverse function, many of which are expressed in sperm but repressed in the zygotes. We have included the data in Fig S3 and dataset1 and commented in the text:

“Genes with diverse functions were associated the hyper DMRs in the hybrid and inbred zygotes versus sperm (**Supplementary Fig. 3c-f; Supplementary Dataset 1**). Most of the genes were lowly expressed or repressed in both sperm and zygotes, while a number of genes were expressed in sperm but repressed in the zygotes (**Supplementary Dataset 1, labeled in red**)”

Line 88: "(Fig. 1)", what part of the panel is this referring to?

Response: It is referring to the Fig. 1a, b, specified now in the text.

“Parental allele-specific methylation was restored during embryogenesis and stably maintained in the hybrids”

Here the authors nicely show that although the DMRs between egg and sperm are closer to the maternal genome in hybrids, they recover the parent-specific profile by the globular stage. I wonder if this is consistent with the activity of the RdDM pathway during embryogenesis? The authors have RNAseq data that can help answer this question.

Response: Thanks for the comments. We have analyzed transcript levels of RdDM genes in zygote and GE (Fig S7)

and commented the data in the text:

“Transcript levels of genes involved in CHH methylation (e.g. *AGO4*, *DCL3*, *DRM2*, and *Pol IV*) were lower in the GE than in the zygote (**Supplementary Fig. 7**).

Figure 3. Label x axis.

Response: Thanks. We have added the x-axis in Fig.3 in the revised version.

“Parental methylation difference was associated with distinct histone modifications”

Figure 4. I suggest including at least in the extended data a representation of the data using box plots, which are more quantitative and less sensitive to bias from individual

loci.

Response: Thanks for the suggestion. We have included boxplots in the Supplementary Fig. 9 in the revised version.

Supplementary Fig.9: Histone modification levels of the egg-sperm DMRs in the parental lines.

“Parental DNA methylation remodeling mirrors parental contribution to zygotic gene expression”

Whether DNA methylation reprogramming after fertilization has an effect on ZGA is still unclear and not discussed by the authors. The fact that DNA methylation of the paternal genome is reprogrammed to reach levels similar to the maternal genome, and yet most genes show predominant maternal expression, suggests that the mechanism by which this maternal-allele preferential expression occurs is independent from DNA methylation, or that it occurs at an earlier time when the zygote still maintains parental asymmetry in DNA methylation.

Response: Thanks for the comments. We have discussed the point in the Discussion section:

“The observations that DNA methylation of the paternal genome was reprogrammed to reach levels similar to the maternal genome, and yet most genes showed predominant maternal expression, suggest that the mechanism by which this maternal-allele preferential expression occurs at an earlier time when the zygote still maintains parental asymmetry in DNA methylation and that the paternal methylation remodeling may contribute to paternal alleles expression in later stages during zygote development.”

Line 179: Not clear. What do they mean with “most of the reads were maternal allelic-specific and about 1.5%~4.1% of the reads was paternal allelic-specific”? I find hard to reconcile this statement with figure 5a where most genes are biallelic (with a level of maternal bias expression) while a small subset of genes shows either maternal (1063) or paternal (28) specific expression. The authors here refer to Extended Data Figure 10a, which shows a Venn diagram showing, I believe, the parental reads for MEGs in zygotes or for the entire transcriptome? Please rephrase this paragraph for clarity. In the Extended Data Figure 10a. include the total number in Venn diagrams.

Response: Thanks for the comment. We recognize that the presentation and the explanation were not sufficiently clear. We have added the numbers of the SNP reads (maternal and paternal) and expressed SNP genes (maternal and paternal) in the figure (Fig S14a) as well as the overlapping SNP genes between the reciprocal hybrid zygotes. We have rewritten the description of the data for clarity.

“we analyzed parental SNP reads (2.66 to 6×10^6) from the reciprocal hybrid zygote transcriptomes and found that most of the reads were of maternal origin and about 1.5%~4.1% of the reads were of paternal origin (**Supplementary Fig. 14a**).”....

“From the SNP reads, we identified 6245 expressed SNP genes (2221 maternal biased, 219 paternal biased) in the MZ zygote and 7116 expressed SNP genes (1666 maternal biased, 262 paternal biased) in the ZM zygote (Supplementary Fig. 14a). Among the

SNP genes, 3765 overlapped in the reciprocal hybrids (Supplementary Fig. 14a), of which 1063 were maternal, 28 genes were paternal (**Figure 5a**). A number of genes were parental sequence-specific genes. The other genes are mostly enriched in maternal reads in either ZM or MZ zygote, as shown by the density plots (**Figure 5a**).”

Line 18: DNA

Response: Thanks. corrected.

Reviewer #3 (Remarks to the Author):

In the manuscript “Maternal-based paternal DNA methylation remodeling in rice zygote”, authors Liu et al investigate changes in DNA methylation that occur in early rice zygotes. They observe that in regions of the genome with differential methylation between egg and sperm, zygotes show more similarity to the egg cell methylation levels resulting from changes in paternal methylation. As observed in the literature previously, DNA methylation patterns are consistent across development, with a major exception being the sperm cell, so remodeling of paternal methylation is expected.

The authors hypothesized that the maternal genome is acting to “control” the remodeling of the paternal genome, however none of the experiments or analyses demonstrated active involvement of the maternal genome. This is important because although the zygote is more correlated with the maternal than paternal epigenome, so is the DNA methylation profile of all vegetative tissue in the plant. Sperm cells have active remodeling of methylation during development, so the change in paternal methylation is necessary to regain the methylation pattern typical of vegetative cells. In order to make the claim that “paternal DNA methylation remodeling” is “maternal-based” as stated in the title and throughout the manuscript, the authors would need to somehow distinguish between paternal methylation reverting to vegetative levels vs. changing to maternal levels. While some cases of trans-acting

epi-alleles exist in plants (i.e. paramutation), DNA methylation is also inherited trans-generationally and many alleles maintain methylation patterns consistent with the genetic origin of that sequence, as demonstrated in this paper as well.

Response: Thank you for the comments. We recognize that the term “control” was inappropriate; we have revised the relevant sentences throughout the MS text.

We understand the reviewer’s concern about the paternal remodeling in the zygote to maternal pattern or just vegetative pattern. The later possibility seems unlikely in the context of hybrid zygotes, as the paternal alleles of the E-S DMRs have different DNA sequence than the maternal line. As advised, we have performed additional analysis to compare methylation levels of the E-S DMRs in sperm, egg, zygote, seedling and panicle of the paternal lines (Fig S6). The sperm methylation of the E-S DMRs (in the hybrid context) is already close to the vegetative (shoot and panicle) tissues of the paternal line. DNA methylations in egg and sperm are differentially remodeled as shown in Fig S1 of this work and in Zhou et al 2021 Mol Plant, which can be also observed in the present Fig S6 (where only the E-S DMRs are considered). The data confirm that in the zygote the paternal methylation of the DMRs is remodeled to match the maternal allele levels rather than the levels in vegetative tissues of the parental line.

Fig. S6: DNA methylation levels of the hybrid parental gametes DMR in paternal reproductive cells and tissues.

We have revised the MS text:

“To distinguish between paternal methylation reverting to vegetative levels vs. changing to maternal levels in the hybrid zygote, we analyzed the methylation levels of the E – S DMRs in sperm, egg, zygote, shoot and panicles of the paternal lines used to produce the three hybrids (ZM, MZ and ZHM). We observed that the methylation levels of the DMRs in sperm were similar to shoot and panicle of the 3 paternal lines (**Supplementary Fig. 6a-c**). DNA methylations in egg and sperm of inbred lines are differentially remodeled (**Supplementary Figure 1**)²⁰. These observations suggested that the paternal alleles of the E – S CG and CHG DMRs were remodeled to match the levels of the maternal alleles rather than to restore to the vegetative levels in the zygote.”

In order to make the claim that DNA methylation remodeling is maternal-based, the authors need to exclude the more likely possibility that DNA methylation is simply being re-established to vegetative levels based on cis-regulatory information coded in the DNA. A cis-regulatory mechanism of DNA methylation reestablishment would also explain why the maternal shift occurs for CG and CHG but not CHH methylation which depends on trans-acting factors. The genotype-dependent methylation patterns are demonstrated in Figure 3, however, the authors label this as parental effects instead of genotype effects. Excluding re-establishment of vegetative methylation could potentially be performed with additional analyses of existing data, however it may require additional experiments to demonstrate persuasively.

Response: We thank the reviewer for the comments. In our reply to the previous comment of the reviewer we have provide data confirming the results (shown in Fig 2) that in the hybrid zygote, the paternal methylation of the E-S DMRs are remodeled to the levels of the maternal alleles (Fig 2). In Fig 3 we show that at Globular Embryo (GE) stage, the parental methylation difference (or E-S DMR) is reestablished. We agree that the reestablishment at GE stage may be based on a cis-regulatory mechanism, as we mentioned in the last part of the discussion section. In the context of hybrid, 'parental allelic difference' is in fact 'DNA sequence difference'. We have changed 'parental allelic-specific' to 'parental allelic or sequence-specific' in the discussion section.

In addition to the major problem of the data not matching the main point of the paper, there are several smaller cases of the text of the manuscript not matching or oversimplifying the results shown on figures. I have listed these examples below.

Line 64 – PCA shows separation by genotype before cell type. Please rewrite to reflect observations.

Response: Thanks for the point. We have rewritten the statement in the revised

version.

“Principal component analysis revealed a high reproducibility of the replicates and a clear difference between the two parental lines”

Line 66 – Boxplots show slightly lower CG for sperm in one genotype but higher CG in the other genotype. CHG is very similar.

Response: Thanks. We have rewritten the statement in the revised version.

“Boxplots indicated that sperm cells showed globally lower CHG methylation (mCHG) than egg cells in TEs”

Line 78 – Only some of the bars show that “more than half of DMRs concerned non-TE regions.

Response: We have revised the statement in the revised version.

“Scanning differentially methylated regions (DMRs, within 50-bp windows with the cutoff of methylation difference at CG > 0.5, CHG > 0.3, and CHH > 0.1, $P < 0.05$) between the gametes and zygotes revealed that about a third or more of the DMRs concerned non-transposable element (non-TE) regions”

Line 87 – Methylation was higher for genes but not TEs.

Response: We have revised the statement in the revised version.

“In the hybrid zygotes, the methylation levels appeared higher than in the male and female gametes, particularly at genic CG and CHG sites”

Line 200 – The figures do not look “overall higher” in all regions, but does look marginally higher in upstream and downstream regions. Error bars or other stats on the metaplots would help clarify if the signal here is significant.

Response: Thanks for the comments. We have rewritten the sentence in the revised version.

“and displayed lower mCHH in egg than in sperm in the upstream region”

Line 206 – What is meant by “comparable numbers”? It looks like fewer MEGs and PEGs than zygotes.

Response: The “comparable numbers” is referring to the numbers of paternal allelic-specific (P-ASEGs: ZM [n = 903], MZ [n = 1150]) or maternal allelic-specific expressed genes (M-ASEGs; ZM [n = 1883], MZ [n = 1680]) in ZM and MZ GE.

We have revised the statement in the revised version.

“similar numbers”

Finally, the authors analyze parent-of-origin biased gene expression and conclude that expression patterns are dynamic and shift between MEGs, PEGs, and non-imprint. However, these calls are only described as discreet data. The problem with this is that imprinting calls are typically based on a small number of allele-specific reads, and a change from, for example, MEG to not imprinted can reflect a real change in allelic expression, a loss of expression of the expressed allele, or most often, it can reflect a subtle change that is no longer called significant without necessarily reflecting the biology. In order to conclude that imprinting is dynamic in these early stages, the authors need to demonstrate that the change in patterns is real and not an artifact of small numbers of reads.

Response: We have deleted the sentence.

Reviewer #1 (Remarks to the Author):

The results and analysis reported in this manuscript constitute a major advance in our understanding of the epigenetic regulation of parent-of-origin contributions to early embryogenesis. I am satisfied with the authors' response to my comments and I recommend publication of this manuscript.

While I was reading the manuscript, I noticed that reference 13 is incomplete, the complete reference should be:

(13) Alaniz-Fabián J, Orozco-Nieto A, Abreu-Goodger C, Gillmor CS. Hybridization alters maternal and paternal genome contributions to early plant embryogenesis. *Development* (2022) 149 (22): dev201025.

Reviewer #3 (Remarks to the Author):

The authors have adequately addressed my concerns.

REVIEWERS' COMMENTS

Reviewer #1 (Remarks to the Author):

The results and analysis reported in this manuscript constitute a major advance in our understanding of the epigenetic regulation of parent-of-origin contributions to early embryogenesis. I am satisfied with the authors' response to my comments and I recommend publication of this manuscript.

While I was reading the manuscript, I noticed that reference 13 is incomplete, the complete reference should be:

(13) Alaniz-Fabián J, Orozco-Nieto A, Abreu-Goodger C, Gillmor CS. Hybridization alters maternal and paternal genome contributions to early plant embryogenesis. *Development* (2022) 149 (22): dev201025.

Response: Thanks, we have completed the reference information.

Reviewer #3 (Remarks to the Author):

The authors have adequately addressed my concerns.

Response: Thank you for the comments!